# Homonyms and context in signalling game with reinforcement learning

**Dorota Lipowska** [1]*, **Adam Lipowski** [2], **António L. Ferreira** [3]

1 Faculty of Modern Languages and Literature, Adam Mickiewicz University in Poznań, Poland, 2 Faculty of Physics, Adam Mickiewicz University in Poznań, Poland, 3 Departamento de Física, I3N, Universidade de Aveiro, Portugal

* lipowska@amu.edu.pl

**Data availability statement:** All relevant data are within the manuscript.

## Abstract

Using multi-agent signalling game with reinforcement learning, we examine the influence of context on the dynamics of homonyms. In our approach, context denotes additional information sent to the receiver, which helps to recognise the signal. Agents in our model select a communicated word or its interpretation with a probability proportional to the power of its weight, which accumulates over previous successful communication attempts ($probability \sim weight^{\alpha}$). The behaviour of the model hinges to some extent on whether this probability depends linearly ($\alpha = 1$) or superlinearly ($\alpha > 1$) on the weight. Numerical as well as analytical results show that contextuality stabilizes homonyms and also affects the overall dynamics of language formation. While in the linear regime, contextuality can hinder the formation of an efficient language, in the superlinear regime—it can even speed up the process. Some aspects of the evolution of homonyms in our model can be understood using a certain urn model. Mathematical analysis demonstrates that in the superlinear regime and in the presence of contextuality, the urn model predicts the existence of polarised-like homonyms, while in the linear regime, only symmetric homonyms can exist. Since there are polarised homonyms in natural languages, our work suggests that the superlinear regime (which could be considered as a manifestation of the so-called Metcalfe's law) may be more appropriate to describe language formation than the linear regime.

## Introduction

A signalling game was proposed more than fifty years ago by David Lewis [1] to explain how communicative behaviour can lead to the emergence of conventions. The game turned out to be very versatile and found numerous applications in philosophy, economics, or evolutionary biology. Since language can be considered a convention shared in a population of communicating individuals, the signalling game also finds some applications in linguistics. Of course, the signalling game is a simplified model, which captures only some aspects of extremely complex language formation processes.

Analysis of the standard signalling game has shown that communicating agents usually arrive at the so-called signalling system. In such a state, for each object presented to the

**Funding:** This study was financially supported by the "institute for nanostructures, nanomodelling and nanofabrication"—associated laboratory LA/P/0037/202 within the scope of the projects UIDB/50025/2020 and UID-P/50025/2020 financed by national funds through the Fundação para a Ciência e Tecnologia/ Ministry of Education, Science and Innovation in the form of an award received by ALF. No additional external funding was received for this study.

**Competing interests:** The authors have declared that no competing interests exist.

*sender*, there is a uniquely assigned word that is communicated to the *hearer*, who subsequently correctly recognises the object presented to the *sender*. Such a one-to-one correspondence between objects and words ensures an efficient communication with perfect information transfer. Numerous studies indicate that even some modifications of the standard signalling game lead to a signalling system [2–5].

Having in mind linguistic applications, we have to notice, however, that language, although being highly efficient, is not a signalling system. In basically every natural language, one can find various ambiguities that break a one-to-one correspondence, which seems to reduce the efficiency of communication. Such reduced efficiency of communication raises some questions about functional properties of language and can be even considered an indication that language did not evolve to be used for communication [6]. However, the very efficiency of communication is a more subtle issue and certain information-theoretic arguments indicate that ambiguities can actually result in a greater communication efficiency [7,8].

Among language ambiguities, homonyms or homophones are known for their stability and prevalence [9,10], although a certain analysis of signalling game indicates that such structures are only transient features, and asymptotically the model will reach a signalling system [11]. However, a signalling game in its standard formulation seems to neglect a number of factors that can shape our languages. Very often the communicated word is not the only information conveyed. Sometimes the additional information received by the *hearer* provides the context, which can be very helpful in correctly recognising the object in question. In some cases, the context alone may be enough (or almost enough) to interpret the message, opening the possibility of ambiguous but still efficient communication. Recently, it was argued that contextuality, as well as some other factors such as the cost of signalling or partial conflicts of agents, can explain the stability of language ambiguities [12,13]. Moreover, certain studies based on the evolutionary signalling game indicate that ambiguous strategies can even outperform signalling system strategies [14–16].

In the present paper, we examine a multi-agent signalling game that takes into account contextuality. Most of our results are obtained using numerical simulations with monitoring a communicative efficiency of the model and possibly emerging homonyms. The agents typically start without any predefined strategies but such strategies, e.g., a signalling system or strategies with ambiguities, emerge spontaneously as a result of local communication acts between agents. Our approach is thus different from some previous studies on a signalling game with contextuality, where some already predefined strategies are analysed [12,14,16]. We also performed simulations with homonymous initial configurations, which enabled us to examine the stability of such structures.

Within a signalling game, one can also study some other lexical ambiguities, for example, synonyms, i.e., two (or more) words interpreted as the same object [11]. In this case, however, contextuality does not seem to matter because the synonyms are interpreted correctly even without context. The problem is more complex with polysemes [17], i.e., words with multiple but related meanings. To examine polysemy within a signalling game, one would have to introduce the measure of relatedness between objects. Such an analysis seems to be interesting but is beyond the scope of the present paper.

To optimize their decisions, agents in our model use reinforcement learning. Namely, they accumulate the payoff from successful communications, which affects their decisions about which word to communicate or how to interpret a communicated word. The reinforcement learning is a general mechanism, which finds numerous applications in physical, economic, and social contexts. In this method, the history of actions plays a crucial role and shapes

the future dynamics [18]. An important ingredient of the reinforcement learning is the relation between the accumulated payoff and selection probabilities. Although the linear relation seems to be the most natural [19,20], other possibilities, i.e., sublinear or superlinear relations were also examined [21,22] and some qualitative differences between these cases were reported. In particular, communication of agents in a signalling game with a superlinear reinforcement learning seems to be more efficient than with the linear one [23].

As we will show in the present manuscript, the properties of the signalling game with contextuality with the linear and with the superlinear reinforcement learning are also different. In particular, we argue that the model with the superlinear reinforcement learning could be more suitable to describe (human) language evolution since in this (superlinear) case, the emerging language is typically more efficient and closer to a signalling system. Moreover, only for the superlinear reinforcement learning, the contextuality stabilizes certain non-symmetric homonyms, which, as we suggest, can be related to the so-called polarised homonyms [24–29]. What is more, the evolution of homonyms in our model, under certain simplifying assumptions, can be mapped into the dynamics of a certain nonlinear urn model. As a result, the conditions for the stability of homonyms (symmetric as well as non-symmetric) can be inferred from the asymptotic solution of the urn model.

Of course, it is desirable to explain why the superlinear rather than the linear reinforcement learning is more suitable to describe human language formation processes. In our opinion, a similar mechanism, known as the so-called Metcalfe's Law [30,31], operates in some economic and marketing settings, although we admit that to firmly establish such a relation would require further analysis.

In our multi-agent model, the emergence of communication strategies, which can be interpreted as linguistic conventions, resembles a spontaneous symmetry breaking, which plays an important role in the physical sciences, for example, in ferromagnetic systems. In the spirit of relatively simple statistical mechanics models, formation of linguistic conventions was also examined in a class of multi-agent systems known as the naming game [32–34]. There are also some other approaches, where various aspects of language formation were analysed in a population of communicating agents [35–37].

## Model

In our model, we examine a population of $N$ agents, which play a variant of the signalling game trying to establish names for $n_o$ objects. Each agent ($A$) for each object ($o$) has an inventory, where it stores $n_w$ words ($i$) with their corresponding weights:

$$w(A)_{i,o} \quad A = 1, \dots, N \quad i = 1, \dots, n_w \quad o = 1, \dots, n_o \tag{1}$$

In an elementary step, two agents are randomly selected: speaker ($S$) and hearer ($H$). The speaker chooses an object ($o$) and from the corresponding inventory selects a word ($i$), with a probability of selection ($p(S)_{i,o}$) proportional to the weight of this word ($w(S)_{i,o}$) raised to a certain power $\alpha$. More precisely,

$$p(S)_{i,o} = \frac{w(S)_{i,o}^{\alpha}}{\sum_{j=1}^{n_w} w(S)_{j,o}^{\alpha}} \tag{2}$$

To interpret the communicated word ($i$), i.e., to select the appropriate object ($o$), the hearer takes into account the weights of the communicated word in all inventories using a similar

form of the probability of selection:

$$p(H)_{i,o} = \frac{w(H)_{i,o}^{\alpha}}{\sum_{q=1}^{n_o} w(H)_{i,q}^{\alpha}} \qquad (3)$$

When the hearer's interpretation (i.e., the object it selects) agrees with the speaker's choice, the reward for their communicative success is that they both increase by unity the weights of the communicated word in their respective inventories, which increases the chances of choosing successful words in future communication attempts (reinforcement learning). To avoid an excessive increase of weights, which would bring agent communications to a standstill, we apply a population renewal, which means that the selected agent with a certain (small) probability $p_r$ does not become a speaker but instead is replaced with a newly created agent with all weights $w(A)_{i,o} = 1$. In the simulations described in our paper, we used $p_r = 0.0001$. The population renewal is not the only way to avoid a standstill as one can use some other techniques such as memory loss or lateral inhibition [38].

To take into account contextuality, we assume that objects are grouped into $n_o/2$ pairs ($n_o$ is even in our simulations). Namely, the $k$-th pair is made of $2k–1$ and $2k$ objects. Moreover, with probability $p_c$ the speaker communicates to the hearer also the context, namely the number of the pair the object chosen by the speaker belongs to. When the context is transferred, the interpretation is much easier because in such a case the hearer has to choose one of the two objects only. Namely, the hearer still uses Eq (3) but restricted to two objects that belong to the pair that was communicated. In our model, the probability $p_c$ is fixed and independent of the object chosen by the speaker. In more realistic applications, it would be natural to expect $p_c$ to depend on the object chosen.

The reinforcement learning that we implement in our signalling game model can be interpreted as a certain (multi-agent) urn model [18], where one adds balls to urns with probabilities determined by the current number of balls in a given urn. The parameter $\alpha$ that appears in the selection probabilities (2) and (3) controls the feedback between the number of successes and selection probabilities. A similar relation was used in the so-called nonlinear urn model [21] and it was shown to have some important consequences. Namely for $\alpha > 1$, the urn model evolves toward a monopolistic state, where most of the balls end up in a certain urn, while the equilibrium distribution of balls appears for $\alpha < 1$. It was argued that at $\alpha = 1$ the model undergoes some kind of phase transitions, where it intermediates between $\alpha > 1$ and $\alpha < 1$ regimes [21]. In the signalling game model, the monopolistic state plays an important role: if a word gains dominant weight (in a given inventory), it will be used most frequently and it will most likely be interpreted correctly. On the other hand, the equilibrium distribution of balls implies the absence of a dominant word, which hinders the development of the efficient communication. Thus, in the context of language formation processes, the regime $\alpha < 1$ is less interesting.

The model described above, but without contextuality ($p_c = 0$), has already been used to examine the emergence of linguistic coherence [23]. It was observed that such a coherence is typically easier to achieve for $\alpha > 1$ than for $\alpha = 1$. However, some other parameters of the model like $n_o$, $n_w$, and $p_r$ also affect the evolution of the model [23].

In the signalling game, agents spontaneously, through mutual interactions, develop a communication system. The most efficient communication occurs in the so-called signalling system (Fig 1a), in which each object is assigned one word, and each word corresponds to one object. However, this is not the only possibility, and sometimes certain language ambiguities can appear such as homonyms (Fig 1b) or synonyms (Fig 1c).

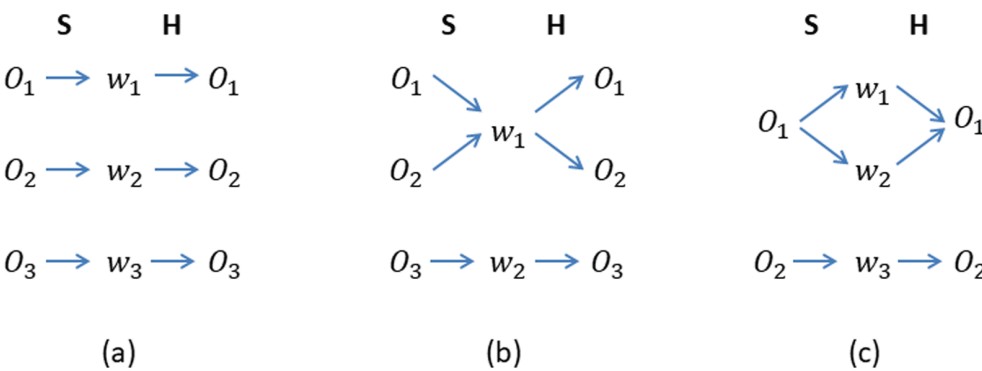

**Fig 1. Signalling system, homonymy, and synonymy in a signalling game.** Speaker (S) assigns a word to the object, communicates it to Hearer (H), who tries to interpret it. (a) Signalling system: most efficient communication occurs when each object is assigned one word, and each word corresponds to one object. (b) Homonymy appears when a word is assigned to two (or more) objects, which reduces the chance of its correct interpretation. (c) Synonymy appears when two (or more) words are assigned to a given object.

For the signalling game model without contextuality ($p_c = 0$), such ambiguities have already been analysed [11]. It has been shown that these structures are usually rather transient and their persistence depends on the parameter $\alpha$. In particular, $\alpha > 1$ seems to make homonyms more stable than synonyms, in accord with some linguistic observations [9,10]. In the present paper, we analyse the dynamics of language ambiguities in the signalling game model that incorporates contextuality.

## Results and discussion

We performed simulations of our signalling game model for values of parameters that we hope are not unrealistic in the context of language formation. We measured the success rate $s$ defined as a ratio of successful communication attempts (in a given unit of time). To measure the level of homonymy in the system, we calculated the homonymy rate $h$ defined as

$$h = 1 - \frac{1}{N n_o} \sum_{A=1}^{N} \sum_{i=1}^{n_w} \frac{\sum_{o=1}^{n_o} w(A)_{i,o}^2}{(\sum_{o=1}^{n_o} w(A)_{i,o})^2} \qquad (4)$$

The parameter $h$ depends on the variance of weights normalized by the square of their average value (for a given agent and a word). For the signalling system, the set of weights peaks on certain objects and such a variance is large, so $h$ equals (or is close to) 0. In such a case, agents establish communication using $n_o$ distinctive words and that is why there is the normalization factor $1/n_o$ in Eq (4). When homonyms (or more general polysems) are present, the weights take more uniform values and $h$ becomes noticeably larger (not exceeding unity).

To examine a time dependence in our model, a unit of time $t = 1$ is defined as $N \cdot n_o$ of elementary steps, i.e., in a unit of time, for each agent, each object is selected on average once by a speaker. Unless specified otherwise, simulations were performed for $N = 300$, $n_o = 30$, and $n_w = 50$. We expect that the presented results are to some extent generic and robust, with respect to changes of the model parameters. In our opinion, however, it is very difficult to provide reliable estimates of these parameters that would correspond to realistic processes of language formation.

## Uniform initial configuration

First, we performed simulations where in the initial configuration all weights $w(A)_{i,o}$ are set to unity. Such a choice does not imprint any structure of the communication (language), which can emerge in our model.

The results of numerical simulations for $\alpha = 1$ and several values of $p_c$ are presented in Figs 2-3. In the absence of contextuality ($p_c = 0$), we can see that for large $t$ the success rate $s$ (Fig 2) reaches a relatively large value while the homonymy rate $h$ drops to a small value. This suggests that agents developed an efficient language that resembles a signalling system, and the bottom panel in Fig 3 supports such an expectation. Indeed, for most objects, there is only one word associated with the object via a large weight (dark pixel) that is not associated with any other object. In the presence of contextuality ($p_c > 0$), additional information transferred to the hearer increases the chance of successful recognition even for homonymous expressions. Thus homonyms quite often are successfully recognised and the homonymy rate $h$ remains large (Fig 2). Somewhat surprisingly, in the presence of contextuality, the success rate $s$ is relatively small (Fig 2). Such a behaviour can be understood when we examine the structure of weights in this case. For example, for $p_c = 0.2$ (middle panel in Fig 3), we can notice a broad distribution of weights, which indicates that the model remains in a state that is unlikely to support an efficient communication and certainly much different from the signalling system. It seems that contextuality supports homonymous expressions and at the same time prevents formation of a signalling system. In the limiting case $p_c = 1$, the success rate $s$ approaches unity but the system remains in the state that is still much different from the signalling system (top panel in Fig 3).

We performed similar calculations for $\alpha = 2$. The large success rate $s$ and small homonymy rate $h$ (Fig 4) suggest that agents reach a signalling system, and the distribution of weights (Fig 5) confirms such a scenario. Moreover, as can be seen in Fig 4, the greater the contextuality, the faster our model seems to reach the signalling system. During the evolution of our model, especially for large $p_c$, some homonyms can form, but these are rather rare events. In fact, it is only for $p_c = 1$ that we can see (Fig 5) that some words have two objects (dark pixels) associated. Of course, when such a word is communicated, the hearer encounters difficulties in interpreting it correctly. Previous analysis of the signalling game model (for $p_c = 0$) [23] has shown that for $\alpha = 2$ there is a stronger tendency to reach the signalling system than for $\alpha = 1$. Our results indicate that such a strong tendency persists even in the presence of contextuality.

## Homonymous initial configuration

In the simulations reported in the previous subsection, agents start the evolution from the uniform initial configuration with all weights set to 1, which means that there is no preference toward formation of a certain signalling (or different than signalling) system. A possible formation of such a state is thus a random process, which can be considered a certain spontaneous symmetry breaking.

During the evolution of the model with the uniform initial configuration, homonyms can form, but these are rather rare events. To make a more systematic analysis of homonyms, it is desirable to start from initial configurations with homonyms already imprinted into them. To perform such simulations, we start from the configuration where the first $n_o/4$ words are homonyms (in our simulations $n_w > n_o/4$). More precisely, for each agent, the $i$-th homonymous word has a large weight (equal to 100) on objects $4i-3$ and $4i-1$, where $i = 1, 2, \ldots n_o/4$. Remaining weights are set to unity, including all weights on non-homonymous objects. In our simulations, we used $n_w = 50$ and $n_o = 32$. Let us notice that when the context is transferred, the hearer knows the pair of successive objects and a homonymous word has a large weight

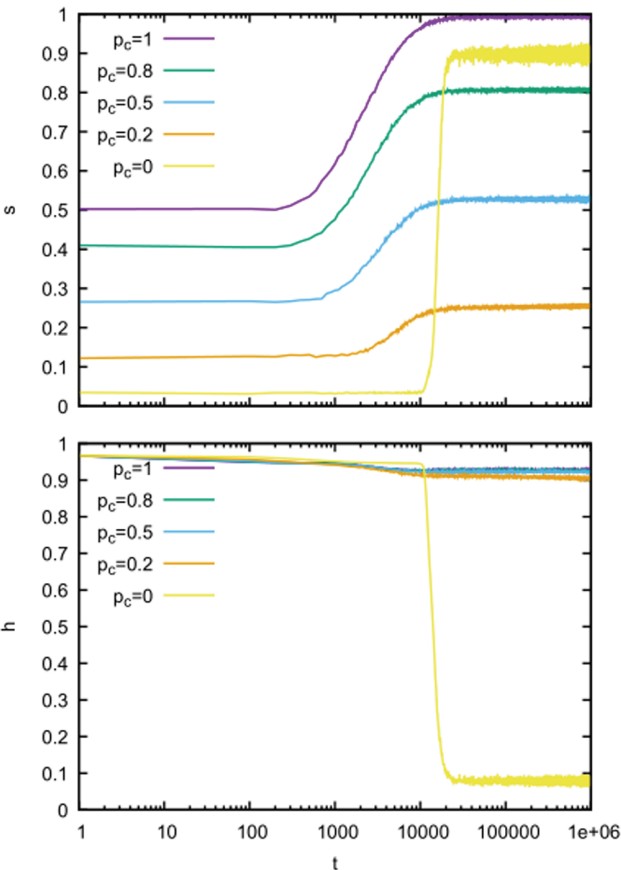

**Fig 2. Time dependence of the success rate $s$ and homonymy rate $h$.** Calculations were made for $N = 300$, $\alpha = 1$, $n_o = 30$, $n_w = 50$, and several values of the context probability $p_c$. All weights in the initial configuration were set to unity.

only on one of these objects. It means that knowing the context, the hearer can (usually) interpret the homonymous word correctly.

For $\alpha = 1$ and in the presence of contextuality, the structure of the language used by the agents remains basically unchanged during the evolution of the model. It means that $n_o/4$ homonyms from the initial configuration persist in the system, as clearly demonstrated in Fig 6 (top and middle panels). The persistence of homonyms also means that the success rate $s$ remains noticeably smaller than 1 (except $p_c = 1$) and diminishes with decreasing $p_c$ (Fig 7). What is more, however, the weights of non-homonymous (even-numbered) objects remain relatively small and agents cannot establish the efficient communication on them. In the absence of contextuality ($p_c = 0$), the initial homonyms disappear (Fig 6, bottom panel) and agents develop a (nearly) signalling system.

Simulations for $\alpha = 2$ show that in the presence of contextuality, homonyms also persist (Fig 8 top and middle panels). However, on a non-homonymous object, agents establish an efficient communication, which results in a larger (than for $\alpha = 1$) success rate $s$ (Fig 9). In the absence of contextuality, similarly to the $\alpha = 1$ case, homonyms disappear and agents develop a signalling system (Fig 8 bottom panel and Fig 9).

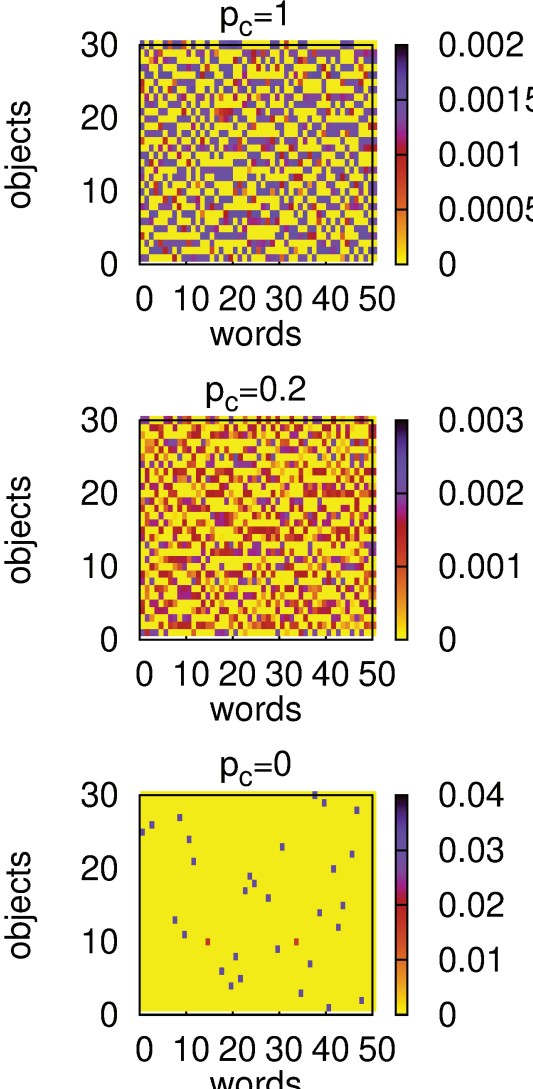

**Fig 3. Heat maps of average and normalized weights** $w_{i,o} = \frac{\sum_{A=1}^{N} w(A)_{i,o}}{\sum_{A=1}^{N} \sum_{i=1}^{n_w} \sum_{o=1}^{n_o} w(A)_{i,o}}$ **at** $t = 10^6$. Calculations were made for $N = 300$, $\alpha = 1$, $n_o = 30$, $n_w = 50$, and several values of the context probability $p_c$. All weights in the initial configuration were set to unity. Only in the absence of contextuality ($p_c = 0$) does the language developed by agents resemble a signalling system with about 30 words with large weights.

## Nonlinear urn model

To understand some aspects of the dynamics, we approximate our signalling game model with a certain urn model. For simplicity, let us consider only one homonymous word H shared between two objects $o_1$ and $o_2$. It means that in the two inventories corresponding to the objects $o_1$ and $o_2$, the weights assigned to the word H dominate, i.e., most likely it is the word H that is communicated whenever the speaker selects $o_1$ or $o_2$. Moreover, we assume that H has negligible weights in inventories for objects different than $o_1$ or $o_2$, i.e., whenever H is communicated, it is (most likely) recognised as $o_1$ or $o_2$. Denoting as $x_1$ and $x_2$ the weights

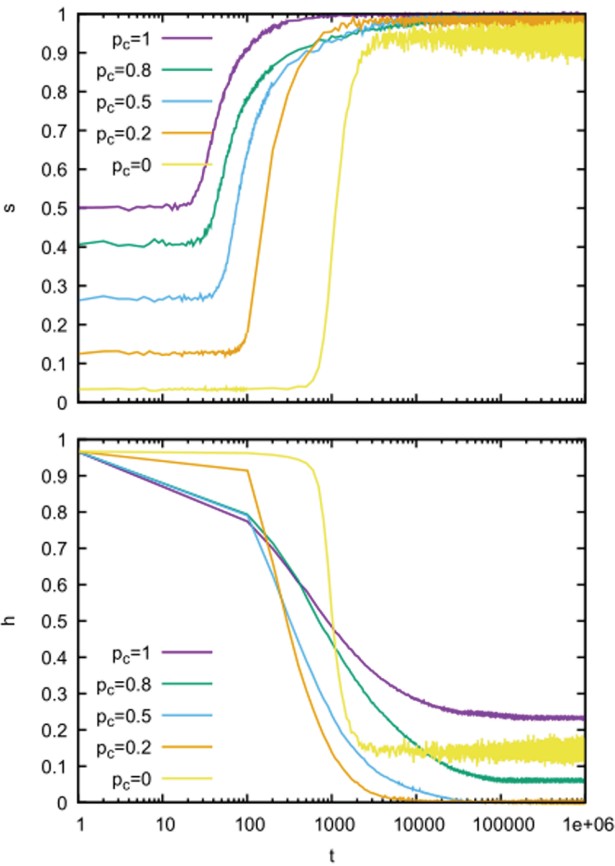

**Fig 4. Time dependence of the success rate $s$ and the homonymy rate $h$.** Calculations were made for $N = 300$, $\alpha = 2$, $n_o = 30$, $n_w = 50$, and several values of the context probability $p_c$. All weights in the initial configuration were set to unity.

of H in inventories corresponding to $o_1$ and $o_2$, respectively, we can write the following evolution equations:

$$x_1(t+1) = x_1(t) + \frac{1}{2}\left[p_c + (1-p_c)\frac{x_1^\alpha(t)}{x_1^\alpha(t) + x_2^\alpha(t)}\right] \tag{5}$$

$$x_2(t+1) = x_2(t) + \frac{1}{2}\left[p_c + (1-p_c)\frac{x_2^\alpha(t)}{x_1^\alpha(t) + x_2^\alpha(t)}\right], \tag{6}$$

where the terms in square brackets are the probabilities of successful recognition of the object selected by the speaker (in such a case, the weight is increased by 1). In a unit of time, the objects $o_1$ and $o_2$ are selected with equal probabilities $\frac{1}{2}$. Moreover, we assume that providing the context (with probability $p_c$) enables the hearer to unambiguously recognise the object. For $p_c = 0$, Eqs (5)-(6) are equivalent to the nonlinear urn model [21]. In such a case, one can argue that asymptotically (i.e., for $t \to \infty$) for $\alpha > 1$, either $x_1$ or $x_2$ dominate and the model is in the monopolistic regime, which in our case implies elimination of the homonymy.

For $p_c > 0$, the analysis of Eqs (5)-(6) is more complicated. Some insight can be obtained if we assume that asymptotically ($t \to \infty$) $x_1$ and $x_2$ increase linearly in time, namely

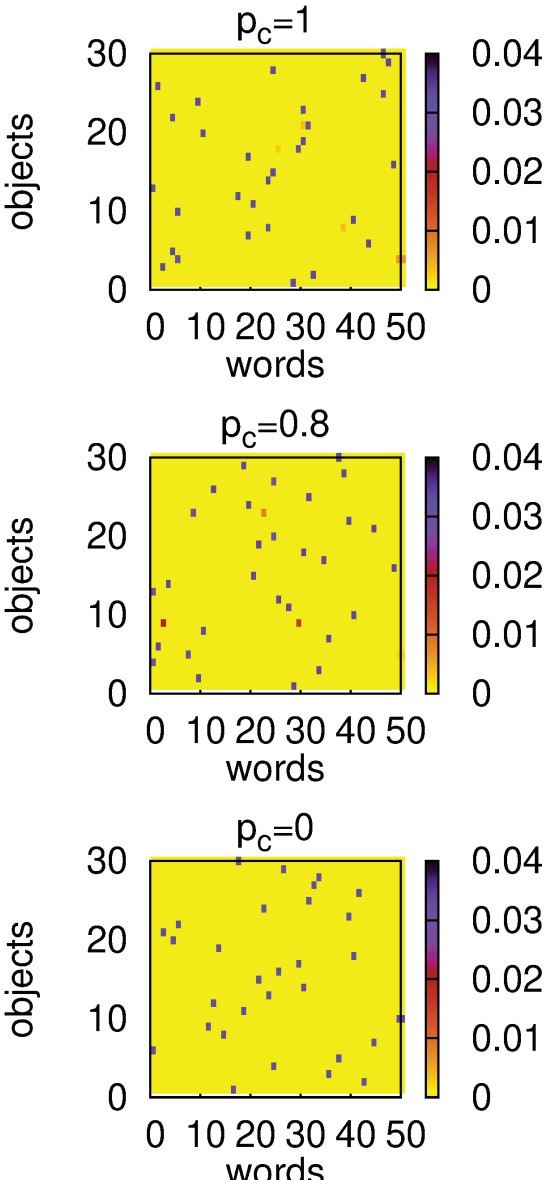

**Fig 5. Heat maps of average and normalized weights $w_{i,o}$ at $t = 10^6$.** The weights $w_{i,o}$ are defined as in the caption of Fig 3. Calculations were made for $N = 300$, $\alpha = 2$, $n_o = 30$, $n_w = 50$, and several values of the context probability $p_c$. All weights in the initial configuration were set to unity. For any context probability $p_c$, the language developed by agents resembles a signalling system.

$$x_1(t) = at, \; x_2(t) = bt. \tag{7}$$

In such a case, after dividing Eq (5) by Eq (6), we obtain

$$\frac{a}{b} = \frac{a^\alpha + p_c b^\alpha}{b^\alpha + p_c a^\alpha} \tag{8}$$

Let us notice that the mutual exchange of the coefficients $a \leftrightarrow b$ leaves Eq (8) unchanged,

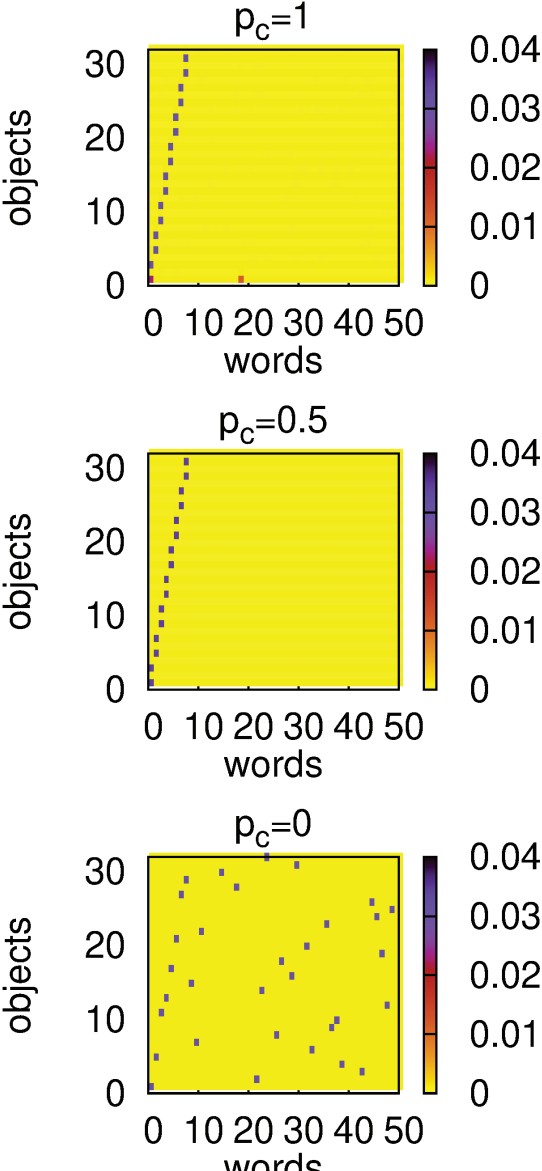

**Fig 6. Heat maps of average and normalized weights $w_{i,o}$ calculated at $t = 10^6$.** The weights $w_{i,o}$ are defined as in the caption of Fig 3. Calculations were made for $N = 300$, $\alpha = 1$, $n_o = 32$, $n_w = 50$, and several values of the context probability $p_c$. The initial configuration contains $n_o/4$ homonyms (see the text). Context ($p_c > 0$) stabilizes the homonyms but hinders the communication about objects that were not strengthened in the initial configuration.

which reflects the symmetry of urns in our model. For $\alpha = 1$ (and $p_c > 0$), the solution of Eq (8) has the form $a = b = (1 + p_c)/4$, which implies persistence of homonymy. When $a = b$, the homonymy is symmetric, which means that the communicated word $H$ is recognised as $o_1$ or $o_2$ with equal probability 1/2.

As we will show in the following, Eq (8) admits also solutions with $a \neq b$, which can be interpreted as non-symmetric homonyms. In general, for $\alpha > 1$, Eq (8) becomes more difficult to analyse. However, in some particular cases, as for example for $\alpha = 2$, relatively simple

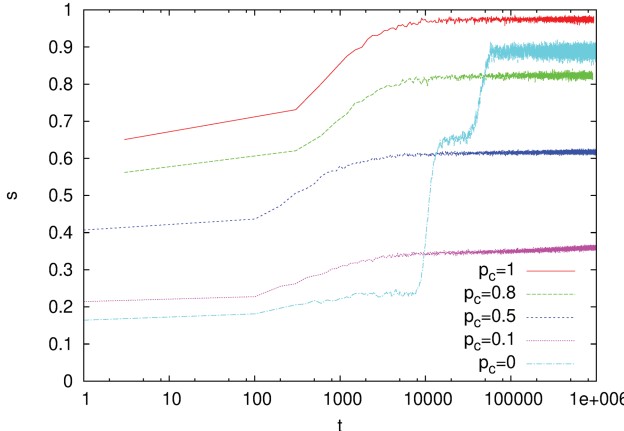

**Fig 7. Time dependence of the success rate $s$.** Calculations were made for $N = 300$, $\alpha = 1$, $n_o = 32$, $n_w = 50$, and several values of the context probability $p_c$. The initial configuration contained $n_o/4$ homonyms (see the text).

analysis can be made. In such a case Eq (8) can be written as

$$p_c z^3 - z^2 + z - p_c = 0 \tag{9}$$

where $z = \frac{a}{b}$. The solutions of Eq (9) have the form

$$z_1 = 1; \; z_2 = \frac{-\sqrt{-3p_c^2 - 2p_c + 1} + 1 - p_c}{2p_c}; \; z_3 = \frac{\sqrt{-3p_c^2 - 2p_c + 1} + 1 - p_c}{2p_c} = z_2^{-1} \tag{10}$$

The solutions $z_2$ and $z_3$ satisfy $z_2 z_3 = 1$, which is of course related to the already mentioned $a \leftrightarrow b$ symmetry.

To check our analysis, and in particular the assumption that the solutions of Eqs (5-6) are indeed asymptotically linear in time $t$, we performed simulations of the urn model that corresponds to Eqs (5)-(6). Namely, starting from $x_1(t = 0) = x_2(t = 0) = 1$, at each time step with probability $\frac{1}{2}\left[p_c + (1 - p_c)\frac{x_i^\alpha(t)}{x_1^\alpha(t) + x_2^\alpha(t)}\right]$, we increased by 1 the weight $x_1(t)$ or $x_2(t)$. After $t = 10^7$ steps of simulations, we plotted the ratio of the weights, either $x_1(t)/x_2(t)$ or $x_2(t)/x_1(t)$, in such a way that it is smaller than (or equal to) 1.

Our results are presented in Fig 10. For $\alpha = 1$, our simulations show that the ratio of weights remains very close to unity, except for very small values of $p_c$. Such a behaviour is in agreement with the analytically predicted solution $a/b = 1$. For $\alpha = 2$, numerical results agree with $z_1 (= 1)$ for $p_c \gtrsim 0.35$ and with $z_{2,3}$ for $p_c \lesssim 0.35$. Based on the linear approximation (7), we analysed the stability of Eqs (5-6) calculating the Jacobian of the 2x2 mapping and its eigenvalues. Leaving aside the details of these elementary calculations, we obtained that the symmetric solution $z_1 (= 1)$ is unstable in the range $0 < p_c < \frac{1}{3}$, and numerical simulations agree with these results. We also performed simulations of our urn model for $\alpha = 1.5$. In this case, we could not find analytically the solution of Eq (8), but it can be easily found numerically. Simulations of the urn model and numerical solution of Eq (8) for $\alpha = 1.5$ are in reasonable agreement, as shown in Fig 10. Let us notice that for $\alpha = 2$ and $p_c = 1/3$, the solutions $z_{2,3}$ are equal to 1. In a more general case of arbitrary $\alpha$, the limiting value $p_c^l$, where solutions

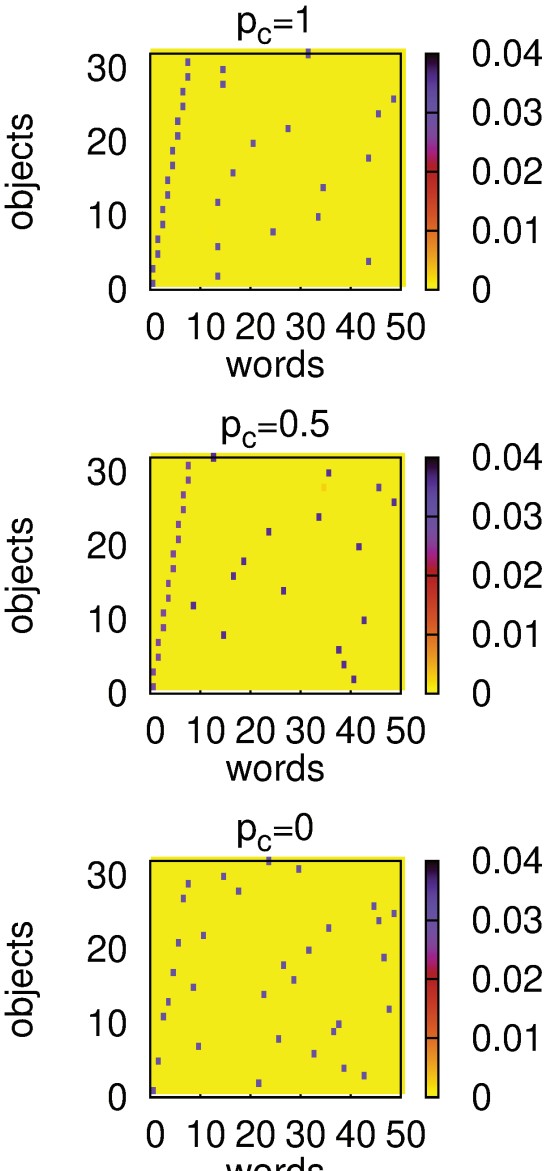

**Fig 8. Heat maps of average and normalized weights** $w_{i,o}$ **calculated at** $t = 10^6$. The weights $w_{i,o}$ are defined as in the caption of Fig 3. Calculations were made for $N = 300$, $\alpha = 2$, $n_o = 32$, $n_w = 50$, and several values of the context probability $p_c$. The initial configuration contained $n_o/4$ homonyms (see the text). Context ($p_c > 0$) stabilizes homonyms, but the emerging language is more efficient than for $\alpha = 1$.

with $a/b \neq 1$ approach the value $a/b = 1$, equals

$$p_c^l = \frac{\alpha - 1}{\alpha + 1} \tag{11}$$

Such a value of $p_c^l$ can be obtained from the Taylor expansion of Eq (8) around $a/b = 1$. Equation (11) shows that $p_c^l$ remains positive for any $\alpha > 1$. It means that for $\alpha > 1$ there is always a range of $p_c$ where non-symmetric homonyms should persist.

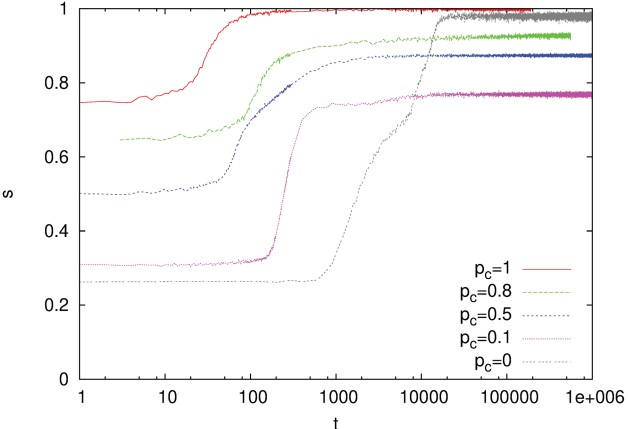

**Fig 9. Time dependence of the success rate $s$.** Calculations were made for $N = 300$, $\alpha = 2$, $n_o = 32$, $n_w = 50$, and several values of the context probability $p_c$. The initial configuration contained $n_o/4$ homonyms (see the text).

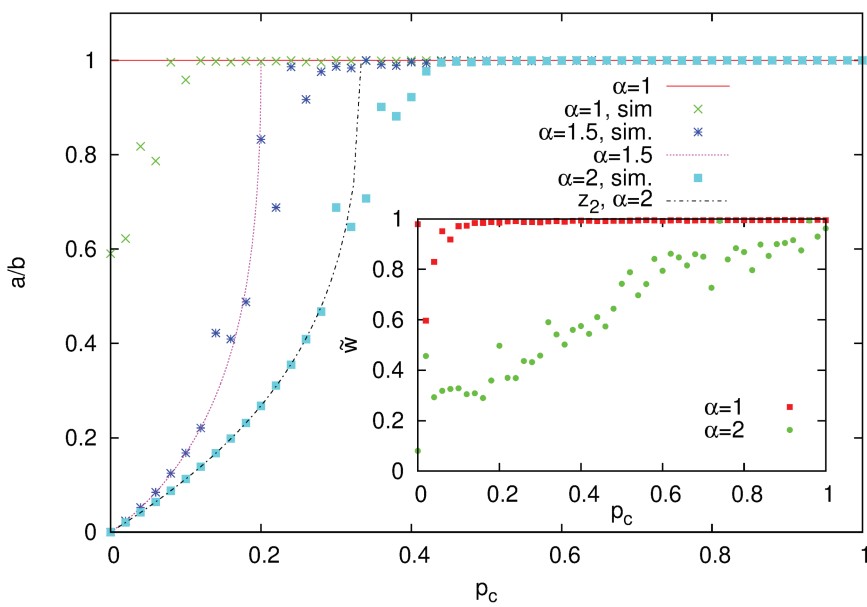

**Fig 10. The ratio of weights $a$ and $b$ as a function of context probability $p_c$ for the urn model (5)-(6).** The line $a/b = 1$ represents the solution $z_1$ in Eq (10) for $\alpha = 2$ as well as the only solution of Eq (8) for $\alpha = 1$. The symmetric homonyms correspond to $a/b = 1$. For $\alpha > 1$ and sufficiently small $p_c$, non-symmetric homonyms persist. The inset shows the ratio of weights $\tilde{w}$ in the signalling game (see text) as a function of $p_c$ calculated for $\alpha = 1$ and 2.

Contrary to the case $\alpha = 1$, where the only solutions are symmetric homonyms (with $a/b = 1$), we obtain that for $\alpha > 1$ in a certain range of $p_c$, our urn model (5)-(6) has non-symmetric solutions with the ratio of weights different than unity. This means that in the signalling game for $\alpha > 1$, we can expect to have non-symmetric homonyms, i.e., words with unequal weights for corresponding objects. In such a case, a homonymous word would be more often interpreted in one way than in the other. To examine such a possibility, we performed simulations of our signalling game model with $n_o = 4$, $n_w = 10$, and $N = 100$. The initial configuration contains a symmetric homonym on objects 1 and 3. Namely, on these objects

the weight of the first word is equal to 100, and of the remaining (nine) ones is equal to 1. For objects 2 and 4, all weights are set to unity. As in the simulations described in the previous subsection, when the context is provided (with probability $p_c$), the pair of objects (1, 2) or (3, 4) is presented to the hearer, which greatly increases the chance of identifying the object (that the speaker had in mind). We performed $10^5$-step simulations, during which we measured the ratio of weights $\tilde{w}$ of the homonymous word on objects 1 and 3. Such a quantity should correspond to the ratio $a/b$ that we examined in the urn model. Results of the numerical simulations for $\alpha = 1$ and 2 and for several values of $p_c$ are shown in the inset of Fig 10. They support the conclusion that for $\alpha = 1$ only symmetric homonyms exist. For $\alpha = 2$, the numerical results are quite noisy but they indicate that in this case non-symmetric homonyms persist, unless $p_c$ is very close to 1.

In the linguistic studies of language ambiguities, one distinguishes the so-called balanced and polarised (unbalanced) homonyms. A homonym is considered balanced if its meanings are used with the same frequency, and polarised otherwise. It should be mentioned, however, that the very definition of the frequency of meaning, as well as its measurement, remains a scientific challenge, and there are numerous works devoted to this subject [39–41]. Research on these frequencies is important, among others, for the study of (impeded or not) cognitive processing of homonyms [24–29]. There are some indications that balanced homonyms are processed more slowly than polarised homonyms [42,43]. This problem is more subtle, however, and some empirical evidence suggests that a relative frequency of homonyms or meaning relatedness also affect the speed of processing [44,45].

Symmetric ($a/b = 1$) and non-symmetric ($a/b \neq 1$) homonyms resemble balanced and polarised homonyms. However, in our signalling game model, we assume that objects (meanings) are presented to the speaker with equal probabilities (frequencies). The asymmetry in the probability of interpretation of a given word is due to the ratio of weights $(a/b)^\alpha$ being different than 1. Our work suggests that it is not only the frequency of usage (i.e., frequency of presentation to the speaker) but also the superlinear reinforcement (with equal frequencies of presentation) that could break the symmetry of homonyms. As we have already mentioned, a nonlinear urn model with $\alpha > 1$ is driven toward the monopolistic solution [21]. Apparently, a similar tendency can be seen for $\alpha > 1$ in the dynamics of homonyms. Let us notice that in the case of homonyms, the prevailing interpretation does not fully dominate and the subordinate interpretation still occurs with a finite probability (because $a/b$ is finite and greater than 0, see Fig 10).

Let us emphasize, however, that non-symmetric homonyms appear in our signalling game only for $\alpha > 1$ and sufficiently small $p_c$ (Fig 10). For $\alpha = 1$, only symmetric homonyms can exist. Linguistic studies indicate that in natural languages both balanced (symmetric) and polarised (non-symmetric) homonyms exist. In our opinion, this may indicate that the signalling game with a superlinear reinforcement ($\alpha > 1$) is more suitable to describe the dynamics of homonymous language ambiguities. Perhaps it would be interesting to extend our signalling game modelling to balanced and polarised versions of polysemous expressions. Such forms are the subject of some recent linguistic research [17,46].

## Conclusions and remarks

Signalling game is a paradigmatic approach to studying the emergence of language through communicative interactions. In our work, we examined a multi-agent signalling game model that takes into account contextuality. It turns out that such additional information transmitted to the listener significantly affects the evolution of the model. Simulations, especially those

where homonyms were imprinted into the initial configurations, show that contextuality stabilizes homonyms. Such a result, observed both for $\alpha = 1$ and $\alpha = 2$, is in accord with some previous works on the role of contextuality in other versions of the signalling game. What is somewhat surprising is the observation that in some cases contextuality hinders the emergence of the efficient communication. Such a behaviour is observed for $\alpha = 1$ both for uniform and homonymous initial configurations. On the other hand, for $\alpha = 2$ contextuality seems to enhance the emergence of the efficient communication.

Analysing a simple urn model that approximates some aspects of the dynamics of our multi-agent model, we demonstrate that contextuality for $\alpha > 1$ (superlinearity) supports the existence of non-symmetric homonyms, which seem to bear some similarity to the so-called polarised homonyms. Our analysis indicates that for $\alpha = 1$ (linearity) only symmetric homonyms can exist. Contextuality is certainly present in human communication and modelling language formation processes should take this factor into account. The fact that both balanced and polarised homonyms appear in natural languages suggests that the signalling game with $\alpha > 1$ provides a better (than $\alpha = 1$) description of human communication. Let us notice that non-symmetric homonyms have not yet been analysed within the framework of the signalling game.

One of the questions arising from our research that may be worth answering is why a superlinear $\alpha > 1$ regime is more appropriate for describing language formation processes, namely, why the probability to select a given word or meaning increases faster than the number of successful communication attempts? Such a strong feedback resembles Metcalfe's Law [30,31], which is expected to hold in some marketing or economic settings. For example, it was demonstrated that the market value of social web services such as Facebook or Tencent seems to increase faster (superlinearly) than the number of its users, in apparent agreement with Metcalfe's Law [47]. In this case, one can argue that each additional user of the service interacts with already existing users, which rapidly increases the network connectivity (and thus its value). It is far from clear to us, whether a similar reasoning can be applied to language formation processes. So far, numerical simulations only suggest that in the $\alpha > 1$ regime our model seems to exhibit some features desirable in the context of language formation. Further research that would clarify the origin of this superlinearity would certainly be advisable.

## Author contributions

**Conceptualization:** Adam Lipowski, Dorota Lipowska, António L. Ferreira.

**Formal analysis:** Dorota Lipowska.

**Investigation:** Adam Lipowski, Dorota Lipowska.

**Methodology:** Adam Lipowski, Dorota Lipowska.

**Software:** Dorota Lipowska.

**Supervision:** Adam Lipowski, Dorota Lipowska, António L. Ferreira.

**Validation:** Adam Lipowski, António L. Ferreira.

**Visualization:** Adam Lipowski.

**Writing – original draft:** Adam Lipowski, Dorota Lipowska.

**Writing – review & editing:** Adam Lipowski, Dorota Lipowska, António L. Ferreira.

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
