## [Decision Letter · Decision Letter 0]

29 Nov 2024

PONE-D-24-44481Homonyms and context in signaling game with reinforcement learningPLOS ONE

Dear Dr. Lipowska,

Thank you for submitting your manuscript to PLOS ONE. After careful consideration, we feel that it has merit but does not fully meet PLOS ONE’s publication criteria as it currently stands. Therefore, we invite you to submit a revised version of the manuscript that addresses the points raised during the review process.

We look forward to receiving your revised manuscript.

Kind regards,

Viacheslav Kovtun, Dr.Sc., Ph.D.

Academic Editor

PLOS ONE

4. We note that your Data Availability Statement is currently as follows: [All relevant data are within the manuscript.]

Reviewers' comments:

Reviewer's Responses to Questions

**Comments to the Author**

1. Is the manuscript technically sound, and do the data support the conclusions?

Reviewer #1: Yes

Reviewer #2: Partly

2. Has the statistical analysis been performed appropriately and rigorously? 

Reviewer #1: No

Reviewer #2: No

3. Have the authors made all data underlying the findings in their manuscript fully available?

Reviewer #1: Yes

Reviewer #2: No

4. Is the manuscript presented in an intelligible fashion and written in standard English?

Reviewer #1: Yes

Reviewer #2: No

5. Review Comments to the Author

Reviewer #1: The paper examines how context in a multi-agent signaling game, enhanced by reinforcement learning, influences the stability and dynamics of homonyms in language formation. By comparing linear and superlinear weight-dependent probabilities, the authors show that contextual cues stabilize homonyms and suggest a superlinear model might better represent real language evolution. Here are my suggestions:

1. The paper claims that introducing contextuality within the signaling game stabilizes homonyms, which is a novel approach, but it could clarify this gap more explicitly. While it notes that previous studies examined signaling games without context or predominantly linear selection probabilities, the paper could benefit from a more direct statement of how these gaps have limited our understanding of language ambiguities.

2. While the paper suggests that the superlinear (α>1) regime aligns better with real-world language evolution, this argument would be strengthened by more empirical or theoretical backing from prior research.

3. The model assumes uniform initial conditions and parameters, limiting generalizability to naturally diverse language environments.

4. The choice of parameter values, while discussed, lacks sufficient justification on why they specifically represent real-world language conditions.

5. Analysis primarily addresses the emergence of stable homonyms but overlooks nuances of actual language ambiguities, such as polysemy and synonymy.

6. Limited exploration of practical implications of superlinear vs. linear regimes for language stability or efficiency in human communication.

Reviewer #2: 1.The manuscript would benefit from improved formatting of formulas and figures to enhance readability and clarity. All equations should be consistently centered. Additionally, multiple similar figures (e.g., time evolution of success rates and homonymy rates for different values of p_c ) could be combined into a single figure with subplots, accompanied by comparative explanations. This approach would make the results more concise and allow readers to better identify trends and relationships. Providing detailed captions that highlight key observations would further improve the visual presentation.

2.While the introduction outlines the basic premise of signaling games and the role of contextuality, the research motivation and its novelty compared to prior works remain unclear. The manuscript should explicitly highlight how this study extends or differs from existing studies on language ambiguities and signaling games, particularly regarding the introduction of superlinear regimes (α>1) and their implications for language formation.

3. The manuscript introduces parameters such as α and p_c but does not provide sufficient justification for their specific values or range. For instance, the significance of α=1 and α>1in reflecting real-world language dynamics needs further explanation. Including a discussion on how these parameters map to observed linguistic phenomena would strengthen the model’s applicability and credibility.

4. The findings on symmetric and nonsymmetric homonyms are intriguing but lack adequate connection to real-world linguistic observations. For example, the distinction between balanced and polarized homonyms is mentioned but not deeply analyzed. The manuscript should provide concrete examples or references to natural language data to validate the claim that the superlinear regime better explains language formation processes. Additionally, discussing the limitations of the current model and potential extensions, such as incorporating more complex contextual features or agent behaviors, would significantly improve the discussion section.

6. PLOS authors have the option to publish the peer review history of their article (what does this mean?). If published, this will include your full peer review and any attached files.

Reviewer #1: No

Reviewer #2: No

---

## [Author Response · Author response to Decision Letter 1]

13 Mar 2025

Dear Editors,

We thank the Reviewers for their critical remarks and comments. We have revised our manuscript according to their suggestions.

Reviewer #1: The paper examines how context in a multi-agent signaling game, enhanced by reinforcement learning, influences the stability and dynamics of homonyms in language formation. By comparing linear and superlinear weight-dependent probabilities, the authors show that contextual cues stabilize homonyms and suggest a superlinear model might better represent real language evolution. Here are my suggestions:

1. The paper claims that introducing contextuality within the signaling game stabilizes homonyms, which is a novel approach, but it could clarify this gap more explicitly. While it notes that previous studies examined signaling games without context or predominantly linear selection probabilities, the paper could benefit from a more direct statement of how these gaps have limited our understanding of language ambiguities.

We have addressed this issue in the modified part of Introduction. In particular, we have mentioned that language ambiguities, as studied using a signaling game without contextuality [11], appear to be transient features only. We have also mentioned that our previous studies [23] demonstrated that with the linear reinforcement, the emerging language is less efficient. When describing the results of the present paper, we have indicated that only for superlinear reinforcement (i) the emerging language is typically more efficient (than for the linear reinforcement), and (ii) in the presence of contextuality, certain non-symmetric homonyms are stable.

2. the paper suggests that the superlinear (α>1) regime aligns better with real-world language evolution, this argument would be strengthened by more empirical or theoretical backing from prior research.

To address this issue, we have modified the final part of the Nonlinear urn model subsection.

3. The model assumes uniform initial conditions and parameters, limiting generalizability to naturally diverse language environments.

In a signaling game with the reinforcement learning, uniform initial conditions imply that there are no predefined strategies. Such strategies, e.g., a signaling system or ambiguous strategies, emerge spontaneously as a result of local communication acts between agents. Thus, our approach does not limit the linguistic diversity of the system. We have modified Introduction to address this issue.

4. The choice of parameter values, while discussed, lacks sufficient justification on why they specifically represent real-world language conditions.

Most of the simulations were done for N=300 (number of agents), n_o=30 (number of objects) and n_w=50 (number of words). We have mentioned in the text that such a choice should not be unrealistic in the context of language formation. Furthermore, we expect that our results are to some extent generic and robust. At the beginning of the Results and Discussion section, we have added that, in our opinion, it is very difficult to provide reliable estimates of these parameters that would correspond to realistic language formation processes.

5. Analysis primarily addresses the emergence of stable homonyms but overlooks nuances of actual language ambiguities, such as polysemy and synonymy.

In Introduction, we have mentioned that, in principle, other ambiguities such as synonyms or polysemes could also appear in the emerging language. For synonyms, however, contextuality does not seem to be important (at least in our approach), since they are correctly interpreted even without context. From the signaling game point of view, polysemes are words with multiple but related meanings, and their analysis within the signaling game is interesting, but probably more complex than that carried out in the present paper for homonyms. We have also mentioned this in Introduction.

6. Limited exploration of practical implications of superlinear vs. linear regimes for language stability or efficiency in human communication.

In the Introduction, we have added a brief discussion of the linear and superlinear regimes as applied to some network formation models, but also in the context of language formation and the signaling game.

Reviewer #2: 1.The manuscript would benefit from improved formatting of formulas and figures to enhance readability and clarity. All equations should be consistently centered. Additionally, multiple similar figures (e.g., time evolution of success rates and homonymy rates for different values of p_c ) could be combined into a single figure with subplots, accompanied by comparative explanations. This approach would make the results more concise and allow readers to better identify trends and relationships. Providing detailed captions that highlight key observations would further improve the visual presentation.

All equations are centered. Figure 2 now combines Figures 2 and 3 from the earlier version of the manuscript, and similarly Figure 4 combines former Figures 5 and 6; these figures show both the success rate and the homonymy rate. Each figure is accompanied by text highlighting key observations.

2.While the introduction outlines the basic premise of signaling games and the role of contextuality, the research motivation and its novelty compared to prior works remain unclear. The manuscript should explicitly highlight how this study extends or differs from existing studies on language ambiguities and signaling games, particularly regarding the introduction of superlinear regimes (α>1) and their implications for language formation.

We have substantially modified the final part of Introduction to better address motivation and novelty of the research. We have also added information on the relations with existing studies, particularly concerning the superlinear enforcing.

3. The manuscript introduces parameters such as α and p_c but does not provide sufficient justification for their specific values or range. For instance, the significance of α=1 and α>1in reflecting real-world language dynamics needs further explanation. Including a discussion on how these parameters map to observed linguistic phenomena would strengthen the model’s applicability and credibility.

To discuss the role of the parameter α, we have modified the final part of the Nonlinear urn model subsection. It is rather difficult to estimate the value of p_c that would correspond to real human communication. In principle, p_c could also depend on the chosen object. In the Model section, we have added:

“In our model, the probability $p_c$ is fixed and independent of the object chosen by the speaker. In more realistic applications, it would be natural to expect $p_c$ to depend on the object chosen.”

4. The findings on symmetric and nonsymmetric homonyms are intriguing but lack adequate connection to real-world linguistic observations. For example, the distinction between balanced and polarized homonyms is mentioned but not deeply analyzed. The manuscript should provide concrete examples or references to natural language data to validate the claim that the superlinear regime better explains language formation processes. Additionally, discussing the limitations of the current model and potential extensions, such as incorporating more complex contextual features or agent behaviors, would significantly improve the discussion section.

We have reformulated the final part of the Nonlinear urn model subsection to better explain the role of the superlinear regime and the relations between our symmetric and non-symmetric homonyms and the balanced and polarised homonyms that have been linguistically studied. We have also mentioned that it would be desirable to extend our approach to polysemous expressions. References [17,46] describing some studies on balanced and polarised polysems have also been added.

---

## [Decision Letter · Decision Letter 1]

26 Mar 2025

Homonyms and context in signaling game with reinforcement learning

PONE-D-24-44481R1

Dear Dr. Lipowska,

We’re pleased to inform you that your manuscript has been judged scientifically suitable for publication and will be formally accepted for publication once it meets all outstanding technical requirements.

Kind regards,

Viacheslav Kovtun, Dr.Sc., Ph.D.

Academic Editor

PLOS ONE

Additional Editor Comments (optional):

Reviewers' comments:

Reviewer's Responses to Questions

**Comments to the Author**

1. If the authors have adequately addressed your comments raised in a previous round of review and you feel that this manuscript is now acceptable for publication, you may indicate that here to bypass the “Comments to the Author” section, enter your conflict of interest statement in the “Confidential to Editor” section, and submit your "Accept" recommendation.

Reviewer #2: All comments have been addressed

2. Is the manuscript technically sound, and do the data support the conclusions?

Reviewer #2: Yes

3. Has the statistical analysis been performed appropriately and rigorously? 

Reviewer #2: Yes

4. Have the authors made all data underlying the findings in their manuscript fully available?

Reviewer #2: Yes

5. Is the manuscript presented in an intelligible fashion and written in standard English?

Reviewer #2: Yes

6. Review Comments to the Author

Reviewer #2: The author has made changes according to the revision opinions and has responded well. I have no futher comments.

7. PLOS authors have the option to publish the peer review history of their article (what does this mean?). If published, this will include your full peer review and any attached files.

Reviewer #2: No

---

## [Editor Report · Acceptance letter]

PONE-D-24-44481R1

PLOS ONE

Dear Dr. Lipowska,

I'm pleased to inform you that your manuscript has been deemed suitable for publication in PLOS ONE. Congratulations! Your manuscript is now being handed over to our production team.

Kind regards,

on behalf of

Prof. Viacheslav Kovtun

Academic Editor

PLOS ONE